# Urinary Metabolite Profiles of Participants with Overweight and Obesity Prescribed a Weight Loss High Fruit and Vegetable Diet: A Single Arm Intervention Study

**DOI:** 10.3390/nu16244358

**Published:** 2024-12-17

**Authors:** Erin D. Clarke, María Gómez-Martín, Jordan Stanford, Ali Yilmaz, Ilyas Ustun, Lisa Wood, Brian Green, Stewart F. Graham, Clare E. Collins

**Affiliations:** 1School of Health Sciences, College of Health, Medicine and Wellbeing, The University of Newcastle, Callaghan, NSW 2308, Australia; erin.clarke@newcastle.edu.au (E.D.C.); maria.gomezmartin@newcastle.edu.au (M.G.-M.); jordan.stanford@newcastle.edu.au (J.S.); 2Hunter Medical Research Institute Food and Nutrition Research Program, Hunter Medical Research Institute, New Lambton Heights, NSW 2305, Australia; lisa.wood@newcastle.edu.au; 3Metabolomics Department, Corewell Health Research Institute, 3811 W. 13 Mile Road, Royal Oak, MI 49546, USA; ali.yilmaz@corewellhealth.org (A.Y.); stewart.graham@corewellhealth.org (S.F.G.); 4Department of Obstetrics and Gynecology, Oakland University William Beaumont School of Medicine, Rochester, MI 48309, USA; 5Jarvis College of Computing and Digital Media, DePaul University, Chicago, IL 60614, USA; iustin@depaul.edu; 6School of Biomedical Sciences and Pharmacy, College of Health, Medicine and Wellbeing, The University of Newcastle, Callaghan, NSW 2308, Australia; 7Institute for Global Food Security, School of Biological Sciences, Queen’s University Belfast, Belfast BT7 1NN, UK; b.green@qub.ac.uk

**Keywords:** urinary metabolites, metabolomics, fruit and vegetables, weight loss

## Abstract

Background/Objectives: Thus far, no studies have examined the relationship between fruit and vegetable (F and V) intake, urinary metabolite quantities, and weight change. Therefore, the aim of the current study was to explore changes in urinary metabolomic profiles during and after a 10-week weight loss intervention where participants were prescribed a high F and V diet (7 servings daily). Methods: Adults with overweight and obesity (*n* = 34) received medical nutrition therapy counselling to increase their F and V intakes to national targets (7 servings a day). Data collection included weight, dietary intake, and urine samples at baseline at week 2 and week 10. Urinary metabolite profiles were quantified using ^1^H NMR spectroscopy. Machine learning statistical approaches were employed to identify novel urine-based metabolite biomarkers associated with high F and V diet patterns at weeks 2 and 10. Metabolic changes appearing in urine in response to diet were quantified using Metabolite Set Enrichment Analysis (MSEA). Results: Energy intake was significantly lower (*p* = 0.02) at week 10 compared with baseline. Total F and V intake was significantly higher at week 2 and week 10 (*p* < 0.05). In total, 123 urinary metabolites were quantified. At week 10, 21 metabolites showed significant changes relative to baseline. Of these, 11 metabolites also significantly changed at week 2. These overlapping metabolites were acetic acid, dimethylamine, choline, fumaric acid, glutamic acid, L-tyrosine, histidine, succinic acid, uracil, histamine, and 2-hydroxyglutarate. Ridge Classifier and Linear Discriminant Analysis provided best prediction accuracy values of 0.96 when metabolite level of baseline was compared to week 10. Conclusions: Urinary metabolites quantified represent potential candidate biomarkers of high F and V intake, associated with a reduction in energy intake. Further studies are needed to validate these findings in larger population studies.

## 1. Introduction

Obesity has become a global health priority due to its increasing prevalence and negative health impacts, including increased morbidity and mortality [1]. This complex and multifactorial disease is defined as excess body fat resulting from long-term excess energy intake [2]. Obesity rates around the world are very concerning. In 2022, 16% of adults worldwide were obese, which means that the prevalence of obesity more than doubled between 1990 and 2022 [2]. The health risks associated with obesity are well-known, including type 2 diabetes (T2DM) [3,4], cardiovascular disease [4,5] and several cancers [4,6].

Greater fruit and vegetable intake has been associated with a reduced risk of obesity and overweight, cardiovascular disease, cancer, T2DM, and all-cause mortality [7,8,9]. Despite these well-documented benefits, global intakes of fruits and vegetables fall short of World Health Organisation recommendations to consume a minimum of 400 g per day [10,11,12]. More specifically in Australia, 96% of the population do not meet national fruit and vegetable recommendations of 7 servings/day [13,14]. Achieving an increase in the consumption of fruit and vegetables would be highly beneficial to the health of the general population.

The STEP study showed that participants who consumed 1, 2, and more than 2 servings of fruits per day, had lower odds of obesity (46% [OR = 0.54; 95% CI: 0.46, 0.64, *p* < 0.001], 29% [OR = 0.71; 95% CI: 0.60, 0.84, *p* < 0.001], and 21% [OR = 0.79; 95% CI: 0.65, 0.95, *p* = 0.014], respectively) compared with those consuming less than one serving. Specifically, individuals consuming 2 servings per day showed 36% lower odds of obesity compared with those consuming less than 2 servings (OR = 0.64; 95% CI: 0.49, 0.84, *p* = 0.002) [15]. These results align with a previous systematic review [16]. Fruit and vegetables also contribute to weight control by being high in nutrients and low in energy, allowing for adequate nutrient intake with reduced calorie consumption [17]. However, meta-analyses results are not consistent as to whether adopting a high fruit and vegetable diet significantly reduces weight [18,19].

Accurately assessing what people eat is essential for investigating relationships between food intake, health, and obesity [20]. Dietary assessment methods such as 24 h dietary recalls, food frequency questionnaires, and food records can be used to obtain detailed information about dietary intake, including fruit and vegetables; however they are at risk of errors [21,22]. Therefore, new tools and methods such as metabolomics, are needed to reflect dietary intake objectively and more accurately and to minimise the differences between studies [23].

Known unbiased measures for diet assessment include data from feeding studies or direct observation and biomarkers [22,24]. Dietary biomarkers are promising instruments for objective dietary assessment; however only a few biomarkers related with diet intake have been identified to date [24,25]. For example, while polyphenols are abundant in plant-based foods such as fruit and vegetables, they are also found in other plant foods, such as tea and coffee, and therefore do not accurately reflect intake only pertaining to fruit and vegetables [26]. This is where advances in technologies and identification of new metabolites through mass spectroscopy may be advantageous as it can help identify metabolites of both diet and metabolism that may be useful candidate biomarkers of diet [23]. Further identification of dietary biomarkers will help to provide a greater understanding of the relationship between dietary intake and health [27,28]. Therefore, the identification of dietary metabolites in biofluids such as urine, may lead to the discovery of objective markers of intake, which could provide valuable information on dietary patterns, nutritional status and the relationship between diet and obesity or overweight. In addition, advancements in the identification of urinary metabolites are promising for having a greater understanding of the dietary metabolome [29,30].

Within the context of a complex diet, several blood metabolites correlate closely with overall fruit and vegetable intake [31,32]. These may represent useful dietary biomarkers, but the identification of similar metabolite biomarkers in urine could be more advantageous since collection is rapid and non-invasive. Furthermore, research is yet to establish whether these metabolite biomarkers that reflect the intake of fruits and vegetables are equally relevant to populations with overweight/obesity seeking to lose weight.

To date, there are no studies that examine the relationship between fruit and vegetable intake, urinary metabolites, and weight change. Thus, the aim of the current study is to answer the following question: “How do the urinary metabolomic profiles compare for participants prior to, during, and after a 10-week weight loss intervention where participants were prescribed a high fruit and vegetable diet?” Further, we wanted to identify classifiers that can distinguish between the different weeks of the fruit and vegetable intake using machine learning.

## 2. Materials and Methods

### 2.1. Study Design

This study is a secondary analysis of a single arm 10-week intervention in which participants were prescribed a high fruit and vegetable diet for weight loss. Data collection occurred at baseline, week 2, and week 10 of the intervention. Measures collected at sessions included anthropometrics, a spot urine sample, demographic characteristics, and dietary intake.

The results from the current study are reported using the CONSORT reporting guidelines for a pilot or feasibility study. Ethics approval for the current study was obtained from The University of Newcastle, Human Research Ethics Committee (H-2013-0315) and the trial was registered with the Australian New Zealand Clinical Trials Registry (ACTRN12620000046909). All participants provided written, informed consent.

Participants enrolled in the intervention received a one-week supply of fruits and vegetables (equivalent to 49 servings (35 servings of vegetables and 14 of fruits) as per the Australian Dietary Guidelines [14]) and dietary counselling from an Accredited Practising Dietitian (E.D.C) at baseline and week 2. These counselling sessions with the dietitian were to provide participants with individualised feedback on their current diet and to provide education and guidance on how to increase their fruit and vegetable intakes for the duration of the 10-week study. The fruit and vegetable boxes provided were the same for all participants and contained a mix of fresh, frozen, and canned options to demonstrate target weekly amounts. Participants who completed the intervention received a $100 gift card to compensate for any additional costs of changing their diet. The full details of the intervention have been published elsewhere [33].

### 2.2. Recruitment

Participants were recruited between August 2017 and March 2018. Briefly, for participants to be included in the intervention, participants had to be aged 18–45 years; have a BMI between 25 and 35 kg/m^2^; have a body fat percentage >20% for males and >30% for females measured via bioelectrical impedance analysis (InBody 720, Seoul, Republic of Korea); be willing to consume a high vegetable and fruit diet; and be interested in reducing their body weight.

To detect a statistically significant total body weight (kg) change (primary outcome) a sample size of 35 participants was required [33]. Of the 43 participants who consented to the intervention, 34 (79%) completed all study measures.

### 2.3. Dietary Intake

Twenty-four-hour recalls were used to assess dietary intake. These were conducted using the multiple-pass Automated Self-Administered 24 h recall system—Australia (ASA-24-AUS) [34,35]. Food and nutrient composition data uses the AUSNUT 2011-13 database [36]. Participants completed each recall approximately three days before and the day of each assessment session. The average of both recalls was taken for each timepoint. Six recalls per participant were collected in total across the 10-week period.

### 2.4. Urine Collection

Fasted spot urine samples were collected at baseline, week 2, and week 10. Further details of how these samples were collected have been reported elsewhere [37].

### 2.5. Urine Analysis

#### 2.5.1. Chemicals

Deuterium oxide (D_2_O, 99.9 atom%D), monobasic potassium phosphate (KH_2_PO_4_, >99.0%), dibasic potassium phosphate (K_2_HPO_4_, >98.0%), 3-(Trimethylsilyl) propionic-2,2,3,3-d4 acid sodium salt (TSP, 98 atom % D, >98.0% (NMR)), and sodium azide (NaN_3_, >99.5%) were purchased from Sigma-Aldrich (St. Louis, MO, USA). Water used throughout the study was purified using a Millipore lab water system (Millipore Sigma, Burlington, MA, USA) equipped with a 0.22 mm filter membrane.

#### 2.5.2. H NMR Sample Preparation and Metabolomics Data Collection

After thawing on ice, a 500 μL aliquot of urine was removed and placed in a 2.0 mL plastic Eppendorf tube. In order to remove the proteins, the samples were centrifuged at 13,000× *g* for 20 min at 4 °C and 360 μL of the supernatant was transferred to a clean 2.0 mL Eppendorf tube. Subsequently, a total of 40 μL phosphate buffer was mixed with 360 mL of supernatant. The samples were vigorously vortexed for 10 s and 200 mL aliquot was transferred into a standard 3 mm thin-walled gNMR Sample Jet tubes (Bruker Biospin, Saugus, MA, USA). The phosphate buffer was prepared as follows, 1.5 M solutions of KH_2_PO_4_ and K_2_HPO_4_ were prepared in D_2_O and mixed using magnetic stirrers at ambient temperature until complete dissolution was obtained. The two solutions, KH_2_PO_4_ and K_2_HPO_4_, were then mixed in a 1:4 (*v*/*v*) ratio, respectively. Then, TSP and NaN_3_ were added to the phosphate buffer at the concentrations of 1.0 mg/mL and 0.13 mg/mL, respectively. The obtained solution was then mixed using magnetic stirrers at the ambient temperature for 10 min and stored overnight to ensure pH equilibration prior to measurement.

All ^1^H NMR spectra of urine samples were randomly recorded at the metabolomics division at Corewell Health Research Institute using a Bruker Ascend HD 600 MHz spectrometer equipped with a 5 mm TCI cryoprobe, automated tuning and matching (ATM) and cooling unit BCU-05, and an automated sample changer (SampleJet, Bruker, Billerica, MA, USA) with sample cooling (278 K) and preheating stations (298 K). Before measurement, each sample was preheated at 298 K for 180 s and kept for 3 min inside the NMR probe head to reach temperature equilibrium of 300 ± 1 K. Then, automatic tuning and matching, lock and shimming were performed using TOPSPIN 3.2. The ^1^H NMR spectra were acquired using a revised version of the standard pulse sequence with water suppression (noesygppr1d) from the Bruker pulse programme library. A total of 256 scans were acquired after 16 dummy scans, and the generated free induction decays (FID) were collected into 64 k data points using a spectral width of 20 ppm. The acquisition time, relaxation delay, and mixing time were set to 5.0, 5.1 and 0.01 s, respectively.

#### 2.5.3. Metabolite Identification and Quantification Using ^1^H NMR Spectroscopy

Prior to spectral analysis, all FIDs were zero-filled to 128 K data points and line broaden by 0.5 Hz. The methyl singlet produced by a known quantity of TSP (1000 μM) was used as an internal standard for chemical shift referencing (set to 0 ppm) and for quantification. All ^1^H NMR spectra were processed and analysed by running Chenomx NMR Suite, a commercially available profiler software designed for 1D NMR data of a mixtures, (v. 9.1, Edmonton, Canada) and normalised to creatinine. All metabolites are reported in μM.

Individual metabolites were grouped according to their biological pathways as reported in the Human Metabolome Database and associated Kyoto Encyclopedia of Genes and Genomes (KEGG) metabolic pathways [38,39,40,41].

### 2.6. Biostatistical and Bioinformatic Analysis

All statistical analysis was undertaken using various statistical packages in R.

#### 2.6.1. Univariate Analysis

Using MetaboAnalyst (v 6.0) [42], a Student’s *t*-test and Mann–Whitney U test were performed for all pair-wise comparisons of in participant characteristics between each timepoint for both parametric and non-parametric distributions, respectively. To account for multiple comparisons, false discovery rates (0.05 < fdr; *q*-values) were calculated. In order to find out whether sample demographics were statistically significantly different, one-way Analysis of Variance analysis (ANOVA) and chi-square tests were conducted using IBM SPSS Statistics toolbox (v. 24.0).

#### 2.6.2. Statistics

*t*-tests were used to determine significant difference in participant characteristics between each timepoint. Spearman correlation was undertaken to assess the relationship between key metabolites identified and fruit or vegetable intakes at each timepoint separately. Bonferroni adjustment was applied. Correlations of <0.2 were classified as weak, 0.2–0.6 moderate, and >0.6 strong as per previous studies [43,44].

#### 2.6.3. Machine Learning Models to Identify Significant Change Between the Weeks

Machine learning models were applied to create classifiers to distinguish the different weeks of the fruit and vegetable intake.

#### 2.6.4. Model Development

To account for any dilution effects, the combined ^1^H NMR metabolomics data were further sum normalised. A metabolite was conservatively excluded if it had missing data in >50% of each group. For all other metabolites, missing measurements were imputed with the knn algorithm for that particular metabolite. Of note, concentration values range over several orders of magnitude both inter- and intra-sample. Therefore, prior to predictive model building, we addressed this by log transforming followed by autoscaling the data. Principal component analysis (PCA) was performed on the pre-processed data to identify any potential outliers using MetaboAnalyst (v.6). The development of the model involved identifying the top 10 features for classification. This approach was chosen to prevent overfitting by not utilising all available features.

Upon feature identification, the number of data points from each class (baseline, week 2, and week 10) was increased to 500 data points using the Synthetic Minority Over-sampling Technique (SMOTE). SMOTE used the actual datapoints and created new points by means of K nearest neighbours. Having more points enables the model to learn better. The artificial data creation is applied only to the training points during the cross-validation. 10-fold cross-validation was used. The results reported are the cross-validation averages obtained.

The performance results for the different week pairs are reported. The performance metrics, along with their full names, are provided in a separate table. The tables are arranged in descending order based on the Area Under the Curve (AUC) value.

#### 2.6.5. Metabolite Set Enrichment Analysis

Metabolite Set Enrichment Analysis (MSEA) was undertaken using MetaboAnalyst (v6.0) [42]. Metabolite names were converted to their respective Human Metabolite Database (HMDB) identifiers, and the raw data were imported in rows. The raw data were subsequently normalised to the sum, log transformed, and auto scaled. The pathway-associated metabolite set was the chosen metabolite library, and all compounds in this library were used. Pathways with a Holm corrected *p*-value less than <0.002 were considered to be statistically significantly different between the various pair-wise comparisons of different timepoints.

## 3. Results

### 3.1. Participant Characteristics, Summary of Urinary Metabolites and Dietary Intake

Participant characteristics are summarised in Table 1. Energy intake was significantly lower (*p* = 0.02), and total fruit intake was significantly higher (*p* = 0.003) at week 10 compared to baseline. Total fruit and vegetable intake was significantly higher at week 2 and week 10 (*p* < 0.05). No other significant differences were identified at any other timepoints (Table 1). In total, 123 urinary metabolites were quantified. Any systematic variation or possible outlier detection was carried out by principal component analysis (PCA). No potential outlier was detected for metabolomics data.

### 3.2. Training Model Accuracy

Training and discovery model performance of logistic regression models were tested and found to be 98% (95% CI 97–99%) accurate for comparisons between baseline and week 2, 99% (95% CI 97–99%) for baseline and week 10, and 94% (95% CI 92–96%) for week 2 and week 10.

### 3.3. Urinary Metabolites

In total, 123 urinary metabolites were quantified (Appendix A). Of these metabolites, there were 48 biological pathways identified, with the primary biological pathways they were involved in being ‘glycine, serine and threonine metabolism’ (*n* = 13, 11%), and ‘urea cycle’ (*n* = 10, 8%). For 32 metabolites (26%) there were no biological pathways recorded in HMDB (Figure 1).

### 3.4. Changes in Urine Metabolites Baseline and Week 2

The 123 quantified metabolites are listed in Appendix A. In addition, Table 2 highlights the metabolites that exhibited statistically significant differences across all timepoints. Daily intake of fruit and vegetables resulted in significant changes (*p* < 0.05) in 36 metabolites (29%) compared with baseline.

The results of the MSEA identified that the top five metabolite groups identified in the urine were related to the citric acid cycle, Warburg effect, amino sugar metabolism, galactose metabolism and arginine and proline metabolism (Figure 2A, Appendix A).

Findings from the multivariate analysis (Figure 2A) show that PC1 explained 15% of the variations and PC2 explained 8% (Figure 2B). The PLS-DA multivariate analysis improved the separation between each time (Figure 2C). Figure 2D highlights the 15 most influential metabolites for the degree of separation between baseline and week 2.

### 3.5. Changes in Urinary Metabolites from Baseline to Week 10

Of the 123 urinary metabolites quantified, 21 metabolites (17%) were significantly different between baseline and week 10 (Appendix A). Of these 21 metabolites, 11 were the same as those identified at week 2 (Table 2). This included acetic acid, dimethylamine, choline, fumaric acid, glutamic acid, L-tyrosine, histidine, succinic acid, uracil, histamine, and 2-hydroxyglutarate. None of these metabolites were significantly correlated with grammes of fruit or vegetable intakes at any time (Appendix A). Overall, there were 16 moderate correlations identified. Of these, one was negatively correlated with fruit intake (glutamic acid [r_s_ = −0.23, *p* = 1.00]) and ten negatively correlated with vegetables, three of which were consistently correlated at baseline and week 10 (glutamic acid [r_s_ = −0.36 to −0.14, *p* = 1.00], histidine [r_s_ = −0.24 to −0.04, *p* = 1.00], and 2-hydroxyglutarate [r_s_ = −0.20 to −0.02, *p* = 1.00]). The remaining five metabolites were positively correlated with fruit intake (fumaric acid [r_s_ = 0.28, *p* = 1.00], glutamic acid [r_s_ = 0.31, *p* = 1.00], L-tyrosine [r_s_ = 0.26, *p* = 1.00], histamine [r_s_ = 0.27, *p* = 1.00], 2-hydroxyglutarate [r_s_ = 0.24, *p* = 1.00].

The results of the MSEA identified that the top five metabolite pathway groups identified in the urine were related to starch and sucrose metabolism, inositol metabolism, Warburg effect, amino sugar metabolism, and aspartate metabolism (Figure 3A, Appendix A). Within these top five metabolite pathways the key metabolites for each were the Warburg effect: Citric acid, D-Glucose, Fumaric acid, Glutamic acid, Malic acid, Lactic acid, Oxoglutaric acid, Pyruvic acid, Succinic acid, ATP, Glutamine, NAD, ADP; Amino sugar metabolism: Acetic acid, Glutamic acid, ADP, ATP, Pyruvic acid; Starch and sucrose metabolism: D-Glucuronic acid, Sucrose, ADP, NAD, ATP; Inositol metabolism: Glucuronic acid, inositol, ADP, NAD, ATP, myo-inositol; Aspartate metabolism: Acetic acid, ADP, ADP, Fumaric acid, Glutamic acid, L-Asparagine, L-Aspartic acid, Oxoglutaric acid, Glutamine, Malonic acid, *N*-Acetyl-L-aspartic acid.

Findings from the multivariate analysis (Figure 3A) show that PC1 explained 14% of the variations and PC2 explained 8% (Figure 3B). Again, the PLS-DA multivariate analysis improved the separation between each time (Figure 3C). Figure 3D highlights the 15 most influential metabolites for the degree of separation between baseline and week 10.

### 3.6. Changes in Urinary Metabolites from Week 2 to Week 10

There were 26 urinary metabolites (21%) that were significantly different between week 2 and week 10 (Appendix A). Only one of these metabolites, dimethylamine, was the same as those identified as significant between baseline and week 2 or 10 (Table 2).

The results of the MSEA identified that the top five metabolite groups identified in the urine were related to starch and sucrose metabolism, pyrimidine metabolism, phenylacetate metabolism, inositol metabolism, and amino sugar metabolism (Figure 4A, Appendix A).

Findings from the multivariate analysis (Figure 4A) show that PC1 explained 13% of the variations and PC2 explained 9% (Figure 4B). The PLS-DA multivariate analysis improved the separation between each time (Figure 4C). Figure 4D highlights the 15 most influential metabolites for the degree of separation between week 10 and week 2.

### 3.7. Machine Learning Outcomes

The results for different pairs of weeks, performance metrics and full names are reported in Appendix A. Tables are ordered by AUC values, from largest to smallest. As anticipated, the week 10 versus baseline comparison exhibited the best performance, followed by week 2 vs. baseline, and week 10 vs. week 2, respectively. An AUC score of 0.96 in the week 10 versus baseline comparison strongly suggests a significant difference between week 10 and baseline.

## 4. Discussion

The current study identified changes in urinary metabolomic profiles in adults with overweight and obesity who completed a weight loss intervention that involved 10 weeks of high fruit and vegetable consumption. Total fruit and vegetable intake significantly increased across the intervention. There were significant differences in metabolite concentrations at each timepoint (baseline, 2 weeks and 10 weeks). Some overlap in the metabolite groups (*n* = 4) and individual urinary metabolites (*n* = 11) was also observed over the course of the 10 weeks. The primary metabolite groups identified were related to amino sugar metabolism, the Warburg effect, starch and sucrose metabolism, and inositol metabolism. The significant individual metabolites identified included acetic acid, dimethylamine, choline, fumaric acid, glutamic acid, L-tyrosine, histidine, succinic acid, uracil, histamine, and 2-hydroxyglutarate. While some of these metabolites may be potential candidate biomarkers of a healthier metabolite signature, based on increase in fruit and vegetable intakes and reduction in total energy as a result of the intervention, further research is required.

Compared to baseline, there were several metabolite groups that characterised the differences over the intervention course. These included metabolites associated with amino sugar metabolism, the Warburg effect, starch and sucrose metabolism, and inositol metabolism. Amino sugar metabolism includes fructose and sucrose metabolism pathways, both of which are modifiable in the diet related to intake of fruits and carbohydrate rich vegetables [45]. Therefore, the increase in amino sugar metabolites over the course of the study is not unexpected; however, further research is required. While the Warburg effect is predominantly associated with cancer metabolism [46], identifying metabolites linked to glycolysis and lactate production in individuals with high fruit and vegetable intake does not necessarily imply the presence of a Warburg-like state [46,47]. Instead, these metabolite patterns may reflect changes in carbohydrate utilisation, gut microbial fermentation, or general metabolic flexibility resulting from dietary components rather than pathological metabolic reprogramming. Further research is required to confirm these observations and clarify their significance. Metabolites of starch and sucrose metabolism are likely to be a result of the dietary intervention. Both starch and sucrose are components of foods rich in carbohydrate and predominately found in fruits and vegetables, as well as grains [45]. Therefore, it is not unusual that metabolites of starch and sucrose metabolism were identified as having changed from baseline. Inositol is found in many plant-derived foods including fruits and vegetables [48]. Consequently, it is also not unexpected that this was a key metabolite group identified throughout the high fruit and vegetable intervention. Additionally, inositol supplementation has been shown to significantly reduce BMI scores in a meta-analysis of clinical trials [49]. Since the intervention focused on weight loss, with an average reduction on body weight of approximately 2 kg (not statistically significant) and a significant reduction in energy intake over 10 weeks [33], higher levels of inositol group metabolites may be partially explained by this reduction in energy intake, which is the key driver of weight loss [50]. Overall, more research into the role of diet, specifically the intake of fruit and vegetables and weight change, along with the presence of the identified metabolite groups, is required. At this time the role of diet and/or energy reduction associated with these four metabolite groups is unknown.

Of the 11 individual metabolites that were significant across all intervention timepoints, almost all (*n* = 9) have been identified as components of fruit and vegetables. Acetic acid is present in fruits and vegetables but particularly fruit [38,51]. In the current study, a significant increase in urinary acetic acid after a high fruit and vegetable diet is not unsurprising, especially as there was a significant increase in fruit intake by the end of the intervention. It is also important to note that acetic acid is a short chain fatty acid produced by bacteria following consumption of fibrous foods fermented in the colon [52,53]. While found commonly in urine, it is not a specific metabolite of fruit and vegetable intakes as it is found commonly in many food sources [54]. Similarly, higher amounts of succinic acid were identified post intervention. Succinic acid is present in both fruit and vegetables [38], but also a gut microbiota-derived metabolite [55], which may confound its relationship as a diet only biomarker. Urinary choline and dimethylamine, have both been reported to increase after the consumption of fruits and vegetables, particularly pulse vegetables [56,57]. This is consistent with the current study which showed significantly higher urinary choline and dimethylamine levels from baseline to 10 weeks after an increase in fruit and vegetables. However, for choline this relationship was not consistent at 2 weeks. Both metabolites have a lack of specificity as a biomarker of fruit and vegetable intakes particularly as the richest sources of choline are from beef, liver, and eggs [58] and dimethylamine is a common component of fish and seafood [59,60,61], which are not plant based foods. Uracil was significantly reduced over the course of the intervention. This metabolite has been detected, but not quantified in vegetables and fruit such as green beans, cauliflower and citrus [38], but is found in highest concentrations in beer [62] which is more likely to explain the negative relationship identified than fruit and vegetables in the current study. Both histamine and its precursor histidine have been associated with fruit and vegetable intakes [38,63,64]. They have been reported in specific vegetables including broccoli and tomato [63,64] which were provided as part of the intervention and therefore may explain why these metabolites were higher post intervention. Fumaric acid and 2-Hydroxyglutarate may be useful candidate biomarkers, but lesser is known about these metabolites in relation to dietary intake except that they are found in small amounts in fruit and vegetables [38]. Overall, a few consistent biomarkers were identified in the current study, of which several are constituents of fruit and vegetables. However, of these metabolites there are other confounding factors such as metabolites influenced by the gut microbiota or intake of other non-plant-based foods that at this time impact the application of these metabolites as biomarkers of fruit and vegetable intakes.

Several metabolites that consistently changed across the intervention have been identified to contribute to obesity or weight loss. Acetic acid has been shown to have anti-obesity effects in mice [53,65,66]. While less is known in humans, there is promising evidence from reviews that show increased production of short chain fatty acids, including acetate, from eating a high fibre fruit and vegetable rich diet can reduce body weight [67]. These findings suggest that acetic acid may play an anti-obesity role in humans as well; however more research is required. Histidine, an essential amino acid, increased across the intervention. Prior research has shown that increased histidine levels play a role in reducing risk factors of metabolic syndrome, including obesity [63]. This may also influence why higher histidine levels were identified during this intervention coinciding with participants weight loss. Gut microbiota-derived metabolites, such as succinic acid, have been identified as a potential therapy to treat obesity and obesity related co-morbidities [55]. Succinic acid receptor (SUCNR1) signalling from macrophages has been suggested as a possible mechanism for resolving obesity related inflammation [55]. Additionally, urinary succinic acid has been shown to have an inverse association with BMI [68], speculated to be related to the effects of gut microbe generated metabolites, diet and tricarboxylic acid (TCA) cycle intermediate impact on the metabolic signature of obesity [68]. While these metabolites may be markers of reduced obesity, there is also a crossover with dietary intake. This adds to the complexity of understanding the impact of diet, metabolites and obesity due to the overlap in metabolic signatures. Overall, the results are promising, suggesting that these metabolites may indicate a healthier metabolic signature even though the nature of their precursors are unknown.

Brain histamine has been shown to play a role in eating behaviour and appetite regulation. Histamine is released as a satiety signal during food consumption [69]. Reduced histamine has been associated with weight gain [70,71]; therefore the increase in histamine over the intervention period may be a result of the intervention effects. This is particularly likely as high fibre, low energy foods such as fruit and vegetables have been shown to help regulate appetite [72,73], with this appetite suppressing role potentially being impacted by the presence of histamine in these foods. Additionally, appetite can be a key driver of weight gain [74]; therefore increases in appetite regulating metabolites, such as histamine, may contribute to the significant decrease in energy consumption post intervention. While there is a causal relationship between higher histamine levels and reduced energy consumption in the current study, no data were collected on appetite, so this relationship needs to be explored further in future studies.

Obesity is an inflammatory condition, characterised by higher levels of inflammatory markers including Interleukin-6 [75]. Elevated levels of circulating inflammatory markers have been associated with metabolic syndrome, cardiovascular disease and T2DM [75]. In the current study, there were significant changes in metabolites which are known to play a role in inflammation. Glutamic acid significantly reduced over the course of the intervention. This metabolite is an amino acid that is a substrate for many pathways, including production of antioxidants and is considered a “fuel for the immune system” [76]. Reduced urinary glutamic acid across the intervention may be partially related to weight loss and associated reduced inflammation, or higher intakes of antioxidants from fruits and vegetables. L-tyrosine is theorised to have beneficial effects on the body including a potential role in the relationship between weight and inflammation with higher L-tyrosine levels associated with reduced weight and reduced inflammation [77]. Therefore, an increase in L-tyrosine from antioxidant rich foods such as bananas, avocado, and plant-based proteins promoted during the intervention may have played a multilayered role in the significant increase in this metabolite across the intervention. Lastly, both histidine and succinic acid are thought to play an anti-inflammatory role in the diet [55,78]. It is theorised that they may contribute to reducing acute obesity related inflammatory responses and may also explain why they both increased across the intervention. However, more research is needed to fully elucidate the mechanisms of action for these metabolites in the complex relationship between diet, obesity, inflammation and metabolic signature.

There are several limitations of the current study that need to be acknowledged. Firstly, spot urine samples were used because it was not feasible to collect 24 h urine samples in the current study. While 24 h urine samples capture urinary metabolites over a longer period of time, spot urine samples have been shown to correlate with 24 h urinary metabolites and can be utilised more easily in a clinical setting. This was a feasibility study, and hence the sample size was not powered for the study outcome and limited further exploratory analysis, e.g., by sex and BMI. Future studies using larger sample sizes could consider further exploring these metabolites by sex or BMI category. Lastly, there was no control group due to the pre-post study design.

## 5. Conclusions

Metabolites identified in the current study represent potential biomarkers of fruit and vegetable intakes and may be useful in studies investigating promotion of fruit and vegetables as a strategy to reduce energy intake as part of weight management interventions and/or prevent the development of chronic disease. Further research in studies with larger samples sizes is required to understand the multidirectional relationship between dietary intake, obesity and metabolic signatures for improving people’s health.

## Figures and Tables

**Figure 1 nutrients-16-04358-f001:**
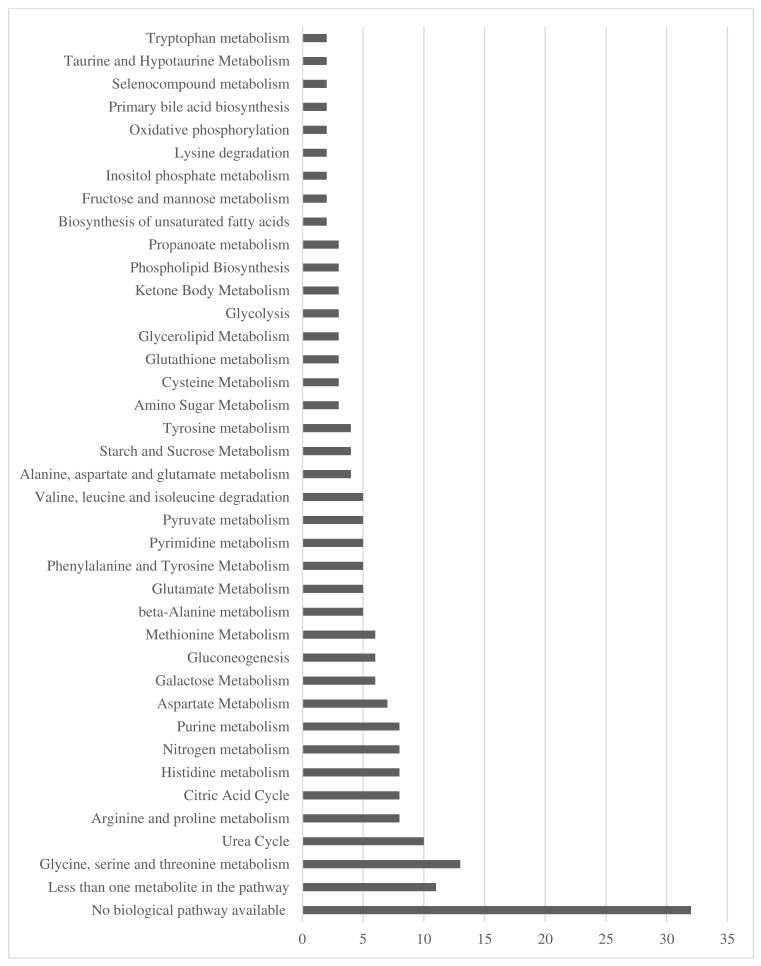
Total number of metabolites classified in each biological pathway.

**Figure 2 nutrients-16-04358-f002:**
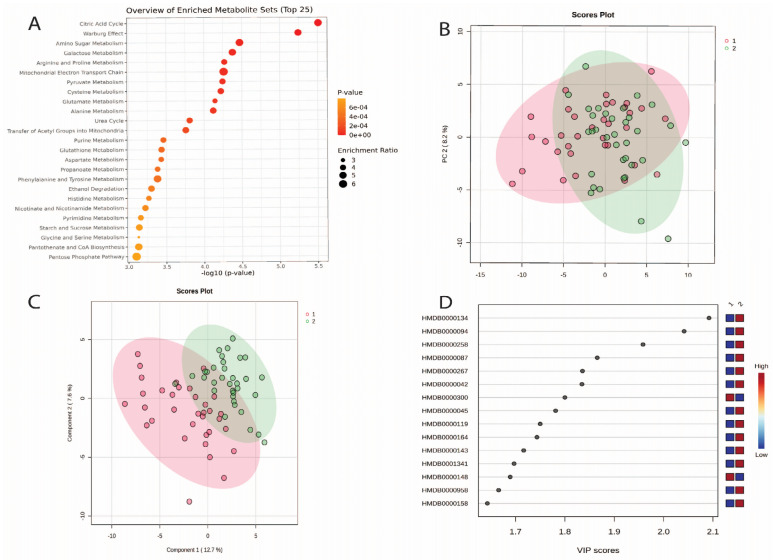
(**A**) Metabolite set enrichment analysis (MSEA) findings for urinary metabolites for baseline and week 2. *p*-values are expressed as scientific notation, these should be interpreted as decimal places. (**B**) Principal Components Analysis (PCA) plot showing group separation. Red dots represent baseline. Green dots represent week 2; (**C**) partial least squares discriminant analysis (PLS-DA) plot showing group separation. Red dots represent baseline. Green dots represent week 2; and (**D**) PLS-DA VIPs highlighting the 15 most important metabolites responsible for observed separation in PLS-DA plots. Metabolite names for Figure 2D from top to bottom: fumaric acid, citric acid, sucrose, dimethylamine, pyroglutamic acid, acetic acid, uracil, adenosine monophosphate, glyoxylic acid, Methylamine, D-Galactose, ADP, glutamic acid, trans-aconitic acid and L-Tyrosine.

**Figure 3 nutrients-16-04358-f003:**
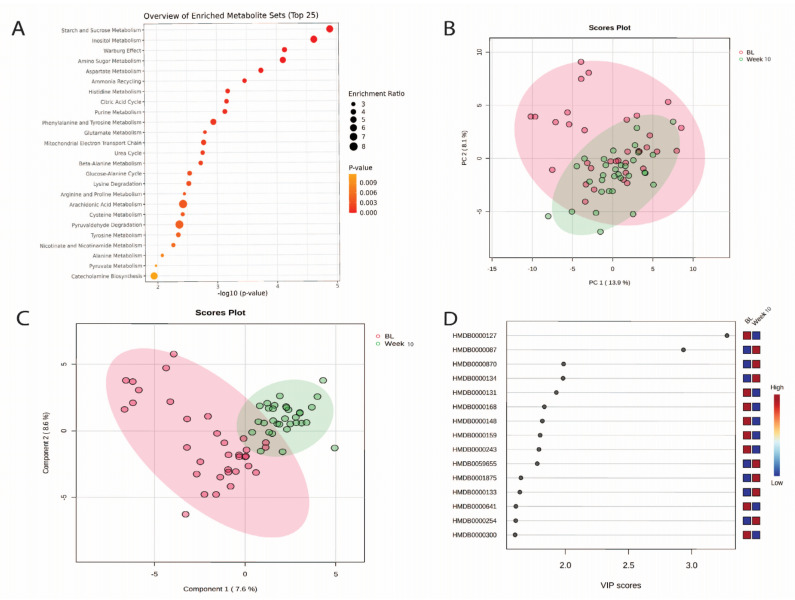
(**A**) Metabolite set enrichment analysis for metabolites for baseline and week 10; (**B**) PCA plot showing group separation. Red dots represent baseline. Green dots represent week 10; (**C**) PLS-DA plot showing group separation. Red dots represent baseline. Green dots represent week 10; and (**D**) PLS-DA VIPs highlight the 15 most important metabolites responsible for separation in PLS-DA plots. Metabolite names for Figure 3D from top to bottom: D-Glucuronic acid, Dimethylamine, Histamine, Fumaric acid, Glycerol, L-Asparagine, Glutamic acid, Phenylalanine, Pyruvic acid, 2-Hydroxyglutarate, Methanol, Guanosine, Glutamine, Succinic acid, and Uracil.

**Figure 4 nutrients-16-04358-f004:**
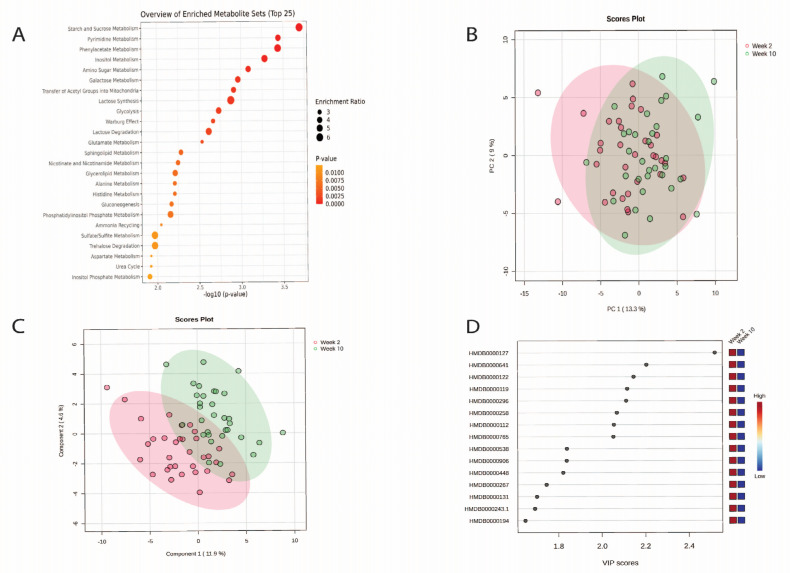
(**A**) Metabolite set enrichment analysis findings for urinary metabolites for week 2 and week 10; (**B**) PCA plot showing group separation. Red dots represent week 2. Green dots represent week 10; (**C**) PLS-DA plot showing group separation. Red dots represent week 2. Green dots represent week 10; and (**D**) PLS-DA VIPs highlighting the 15 most important metabolites responsible for observed separation in PLS-DA plots. Metabolite names for Figure 4D from top to bottom: D-Glucuronic acid, Glutamine, D-Glucose, Glyoxylic acid, Uridine, Sucrose, gamma-Aminobutyric acid, Mannitol, Adenosine triphosphate, Trimethylamine, Adipic acid, Pyroglutamic acid, Glycerol, Pyruvic acid and Anserine.

**Table 1 nutrients-16-04358-t001:** Demographic characteristics and dietary intake of participants (*n* = 34).

Characteristics	Baseline Mean ± SD	Week 2 Mean ± SD	Week 10 Mean ± SD
Age (years)	33.9 ± 8.3	-	-
Sex (*n*, % female)	18 (52.3%)	-	-
Smoking status (*n*, % smokers)	1 (2.9%)	-	-
Weight (kg)	83.6 ± 13.0	82.9 ± 12.8	81.0 ± 11.5
BMI (kg/m^2^)	28.9 ± 1.9	28.6 ± 1.9	28.1 ± 1.7
Energy (kJ/day)	9961.5 ± 4269.4	9774.8 ± 9600.0	7792.1 ± 3236.7 *
Fruit (grammes/day)	157.5 ± 133.5	224.9 ± 151.5	270.2 ± 157.3 *
Vegetables (grammes/day)	333.0 ± 223.7	427.1 ± 318.2	437.2 ± 433.7
Total Fruit and Vegetables (grammes/day)	490.5 ± 267.7	651.9 ± 374.7 *	707.4 ± 469.5 *

* *p* < 0.05 showing statistically significant change from baseline. Differences analysed using *t*-tests.

**Table 2 nutrients-16-04358-t002:** Statistically significant metabolites at all timepoints.

Metabolite Name	Baseline	Week 2	Week 10	Change Baseline vs. Week 2	Change Baseline vs. Week 10	Change Week 2 vs. Week 10
Acetic acid	76.89	130.61	106.23	53.73	26.92	−19.74
	(58.05)	(137.34)	(66.86)	(129.03) *	(71.05) *	(132.60)
Dimethylamine	60.85	199.47	282.46	138.61	226.09	113.48
	(154.02)	(273.16)	(274.65)	(330.64) *	(300.78) *	(250.24) *
Choline	28.61	20.20	33.32	−8.41	3.57	14.83
	(34.12)	(22.06)	(76.41)	(29.04) *	(73.82) *	(71.77)
Fumaric acid	171.13	279.58	283.48	108.45	104.76	20.08
	(176.57)	(215.40)	(220.37)	(229.01) *	(251.33) *	(193.11)
Glutamic acid	256.90	178.90	156.61	−78.00	−107.74	−11.62
	(312.19)	(201.65)	(127.75)	(372.24) *	(349.37) *	(160.98)
L-tyrosine	314.26	533.01	471.12	218.75	142.33	2.79
	(309.82)	(651.49)	(490.67)	(642.76) *	(582.79) *	(712.35)
Histidine	1308.04	2215.59	2154.74	907.55	806.09	249.87
	(1363.49)	(2685.40)	(2133.23)	(2624.58) *	(1666.04) *	(2618.68)
Succinic acid	50.47	82.25	87.64	31.78	35.22	10.80
	(49.86)	(73.00)	(79.98)	(76.59) *	(66.33) *	(87.36)
Uracil	95.61	61.57	68.81	−34.03	−28.76	12.16
	(91.38)	(67.85)	(50.59)	(108.04) *	(94.48) *	(60.82)
Histamine	262.46	397.61	608.75	135.16	331.48	221.90
	(402.60)	(496.59)	(759.52)	(618.69) *	(806.95) *	(763.01)
2-hydroxyglutarate	254.93	562.82	379.36	307.89	114.97	−146.39
	(272.00)	(855.46)	(336.66)	(742.99) *	(316.03) *	(809.86)

All variables reported as mean (SD), with concentrations expressed in μM. * *p*-value <0.05.

## Data Availability

Data may be obtained from the authors upon reasonable request. The data are not publicly available due to ethical reasons.

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
