# Peer review of "Urinary Metabolite Profiles of Participants with Overweight and Obesity Prescribed a Weight Loss High Fruit and Vegetable Diet: A Single Arm Intervention Study"

_nutrients, 2024, doi:10.3390/nu16244358_

Round 1
Reviewer 1 Report
Comments and Suggestions for Authors
Overall, although the objective of the article is relevant and interesting, I consider that the manuscript provides limited value due to the lack of key data to support its content. In addition, multiple spelling errors and oversights are noted, such as the absence of units in several sections, which affects the clarity of the text. Likewise, the information could benefit from better organization and avoid redundancies, as the current structure hinders a clear and concise presentation of the findings.
The following are some points that, in my opinion, the authors should consider for future revisions or work..
1. They did not measure phenolic metabolites or carotenes. Since fruits and vegetables are rich in them, they could be better biomarkers.
2. Section 3.2. Training model accuracy: In line 251 reference is made to week 5, however according to the objective only baseline, week 2 and week 10 are included in this study.
3. In what units are the urinary metabolites expressed in Supplementary Tables 1 and 3?. In addition, I believe that it is better to present a table showing the concentrations of the 123 metabolites at the different times, including the p values derived from comparisons between week 2 and week 10 with respect to the baseline value. Right now, you present two tables, you could unify the names and identifiers in table 1 and in table 2 do what I mentioned above with the concentrations. Also, you should unify how many decimals you present, it's all so confusing. Avoid putting information that you do not mention later in the text.
4. Another example of missing information is that it says “In total, 123 urinary metabolites were quantified” but you do not say in which tables they are, nor to which families they belong, etc. It is so terse. In my opinion, there should be a section where only the metabolites are discussed and within it there should be sub-sections: Changes in urine metabolites Baseline & Week 2 and Changes in urine metabolites Baseline & Week 10
5. Another example of missing information is that it says “Of the 123 quantified metabolites, 36 metabolites significantly changed between baseline and week 2 (p<0.05; refer to Supplementary Table 1)”. However, you do not detail further: what percentage of the total metabolites they represent, or make a figure with these metabolites to present the change. Something to help the reader understand the results. Same for week 10 vs baseline.
6. I would suggest moving Table 3 to the supplementary material. Additionally, it is important to indicate which correlations were significant. Furthermore, what is the explanation for the very low correlations (<0.4)?
7. Table 2 presents the changes observed at two time points. However, I believe it would be more effective to display this information in a graph to better illustrate the concentrations (changes), as the current format of the table provides limited information.
8. Another example of missing information is the following statement: *"The results of the MSEA identified that the top five metabolite groups found in the urine were related to starch and sucrose metabolism, inositol metabolism, Warburg effect, amino sugar metabolism, and aspartate metabolism (Figure 2A, Supplementary Table 4)."* Why not include the names of the metabolites? Don’t hesitate to mention them!.
9. Avoid leaving page 5 blank
10. There are also occurrences of typographical errors with the units. sometimes they are spaced from the numbers or sometimes together. sometimes they are not in superscript, and so on.
Author Response
Dear Reviewer 1, thank you for your feedback. Please see attached our point-by-point responses.

Reviewer 2 Report
Comments and Suggestions for Authors
Please see the attachment.

Author Response
Dear Reviewer 2, Thank you for your feedback on our manuscript. Please see attached out point-by-point responses.

Round 2
Reviewer 1 Report
Comments and Suggestions for Authors
Dear Authors,
Please find the revisions to this article. I appreciate the efforts made by the authors to improve the manuscript. However, I believe there are still several points that require further revision before the article can be considered for publication.

Author Response
Thank you for your feedback, please see attached our point-by-point responses.

Round 3
Reviewer 1 Report
Comments and Suggestions for Authors
Thanks to the authors for their efforts to improve this article. In my opinion this article can be published in its current version.